Microbiology **Spectrum**

🔓 | **Open Peer Review** | Antimicrobial Chemotherapy | Observation

# Amino acid 17 in QRDR of Gyrase A plays a key role in fluoroquinolones susceptibility in mycobacteria

Shuai Wang,[1,2,3,4] Jingran Zhang,[1,3,4,5] H. M. Adnan Hameed,[1,2,3,4] Jie Ding,[1,3,4,6] Ping Guan,[7] Xiange Fang,[1,2,3,4] Jiacong Peng,[3,8] Biyi Su,[7] Shangming Ma,[7] Yaoju Tan,[7] Gregory M. Cook,[9,10] Guoliang Zhang,[11] Yongping Lin,[3,8] Nanshan Zhong,[3,8,9] Jinxing Hu,[7] Jianxiong Liu,[7] Tianyu Zhang[1,2,3,4]

**ABSTRACT** The polymorphism at amino acid 17 of quinolone resistance-determining region of GyrA has been stated with a potential role in fluoroquinolone susceptibility in different mycobacterial species. However, no study has provided dependable evidence so far. Here, we verified that gene-edited *Mycobacterium abscessus* mutants bearing Ser/Gly at this position were more susceptible to fluoroquinolones than their parent strain and the revertant that supports mycobacteria containing Ser/Gly at this position were more susceptible to fluoroquinolones than those containing Ala.

**IMPORTANCE** Fluoroquinolones (FQs) play a key role in the treatment regimens against tuberculosis and non-tuberculous mycobacterial infections. However, there are significant differences in the sensitivities of different mycobacteria to FQs. In this study, we proved that this is associated with the polymorphism at amino acid 17 of quinolone resistance-determining region of Gyrase A by gene editing. This is the first study using CRISPR-associated recombination for gene editing in *Mycobacterium abscessus* to underscore the contribution of the amino acid substitutions in GyrA to FQ susceptibilities in mycobacteria.

**KEYWORDS** fluoroquinolone, mycobacteria, intrinsic resistance, gene editing, *Mycobacterium abscessus*

Fluoroquinolones (FQs) are very important and extensively used class of synthetic antibacterial agents that are recommended for drug-resistant *Mycobacterium tuberculosis* and some non-tuberculous mycobacteria such as macrolide-resistant *Mycobacterium abscessus* complex (1–3). Moxifloxacin (MOX) is known to be an effective FQ and a key component of the new first-line regimen which can shorten the treatment duration of drug-sensitive tuberculosis from 6 months to 4 months (4). The target of the FQs in mycobacteria is type II topoisomerase, which consists of two subunits, GyrA and GyrB, that form the catalytically active $A_2B_2$ heterotetrameric structure (5). In mycobacteria, the FQs can interact with the cleaved DNA together with the GyrA and GyrB proteins to stabilize a cleavage complex and inhibit the religation of the cleaved DNA which potentially results in lethal double-strand DNA breaks in the genome (6). Mutations in quinolone resistance-determining region (QRDR) of *gyrA* and *gyrB* genes have been proven to lead to FQ resistance (7, 8).

Interestingly, most mycobacteria are intrinsically less susceptible to FQs than other bacteria, such as *Escherichia coli*, and the levels of susceptibility to FQs differ markedly for different mycobacterial species (9). Previous studies found the amino acid substitution at position 17 in QRDR[GyrA] (83 in *E. coli*; 90 in *M. tuberculosis*; 92 in *M. abscessus* of GyrA) may be involved in the FQ susceptibility by analyzing the sequences of the QRDR in GyrA and GyrB in different mycobacterial species as well as other bacteria (9, 10). The

Address correspondence to Jinxing Hu, hujinxing2000@163.com, Jianxiong Liu, ljxer64@qq.com, or Tianyu Zhang, zhang_tianyu@gibh.ac.cn.

Shuai Wang and Jingran Zhang contributed equally to this article. Author order was determined in order of decreasing seniority.

The authors declare no conflict of interest.

See the funding table on p. 4.

presence of an Ala at position 17 of QRDR$^{GyrA}$ in most of the mycobacterial species (such as *M. tuberculosis* and *M. abscessus*) and a Ser in the three other mycobacterial species (*Mycobacterium peregrinum, Mycobacterium fortuitum,* and *Mycobacterium aurum*) or *E. coli* associated with the minimal inhibitory concentrations (MICs) of quinolones, suggesting this amino acid residue might be a crucial determinant of different susceptibilities to quinolones among mycobacteria (Fig. 1A) (9, 10). However, to date, no direct molecular experimental evidence supports this hypothesis. The success of mycobacteria gene editing with the development of CRISPR (11) allowed us to apply this new tool for exploring the correlation between FQ susceptibility patterns and the amino acid substitutions in mycobacteria.

*M. abscessus* is a rapidly growing non-tuberculous mycobacterium responsible for a wide variety of human diseases, including chronic pulmonary diseases and several

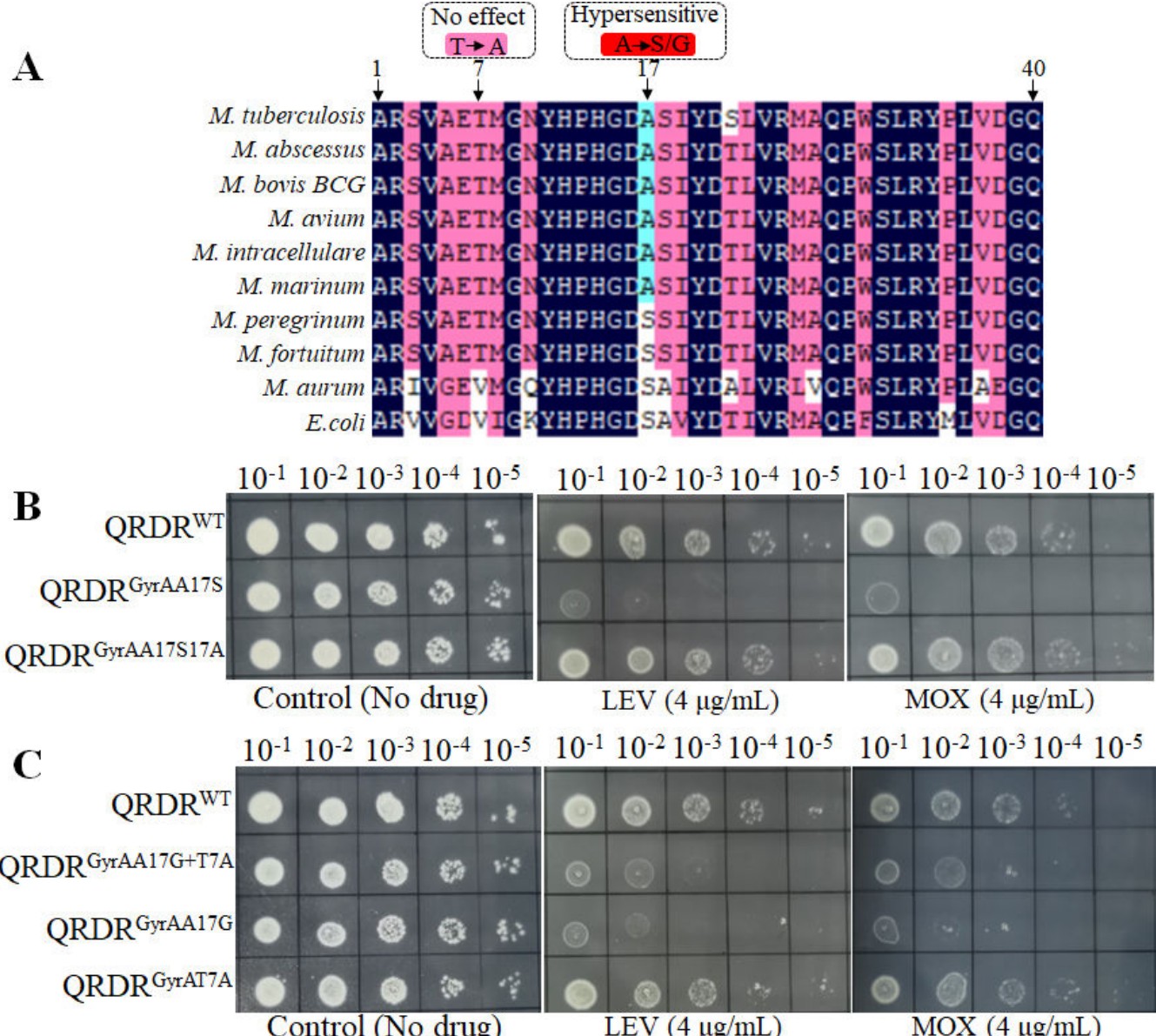

**FIG 1** The amino acid 17 in QRDR of GyrA contributes to intrinsic susceptibility of FQs in mycobacteria. (A) Alignment of the peptide sequences of the QRDR of GyrA from mycobacterial species and *E. coli*. (B and C) Drug susceptibility of different *M. abscessus* strains to MOX and levofloxacin. Tenfold serial dilutions of *M. abscessus* strains grown to OD$_{600}$ of 0.7 were spotted on Middlebrook 7H10 containing indicated concentrations of antibiotics. Plates were incubated for 3 days. Representative data from three independent experiments are shown.

extrapulmonary diseases such as soft tissue, skin, and central nervous system infections (12). These infections are difficult to treat with the standard antibacterial therapy due to their high-level intrinsic resistance to most antibiotics (13). Here, we used *M. abscessus* as a model organism for studying the correlation between FQ susceptibility patterns and amino acid sequences of QRDR$^{GyrA}$ in mycobacteria. To elucidate the contribution of the amino acid substitution located at position 17 of QRDR$^{GyrA}$, the GyrA was edited to alter the Ala to Ser at position 17 of QRDR$^{GyrA}$ (QRDR$^{GyrAA17S}$) in *M. abscessus* using CRISPR-associated recombineering as described previously for other mycobacteria (11). The primers used in this study are listed in Table S1. Interestingly, QRDR$^{GyrAA17S}$ exhibited a markedly enhanced sensitivity to levofloxacin (LEV) and MOX compared to its parent *M. abscessus* strain (QRDR$^{WT}$) as shown in Fig. 1B and Table 1. The MICs of both LEV and MOX to the QRDR$^{GyrAA17S}$ were 1/4 of that of Mab$^{WT}$ (Table 1), whereas QRDR$^{GyrAA17S17A}$, the GyrA of which was edited to alter the Ser back to Ala at position 17 of QRDR$^{GyrA}$ in QRDR$^{GyrAA17S}$, recovered the FQ resistance level to that of QRDR$^{WT}$ (Fig. 1B and Table 1), thus confirming that Ala17 in QRDR$^{GyrA}$ is critical for the FQs resistance in *M. abscessus*. Susceptibility of QRDR$^{GyrAA17S}$ to other two types of non-FQ antibiotics remained unchanged (Table 1), indicating that the substitution of this amino acid specifically affects the LEV and MOX susceptibility. Additionally, Aubry et al. found that some *M. tuberculosis* isolates bearing a combination of T80A and A90G (Thr7Ala and Ala17Gly in QRDR$^{GyrA}$) substitutions were hypersusceptible to ofloxacin (14). Therefore, we also constructed a *M. abscessus* mutant (QRDR$^{GyrAA17G+T7A}$) containing Thr7Ala and Ala17Gly in QRDR$^{GyrA}$ double mutations in GyrA. Similar to QRDR$^{GyrAA17S}$, QRDR$^{GyrAA17G+T7A}$ also showed the significantly increased sensitivity to LEV and MOX (Fig. 1C and Table 1). To further investigate the contribution of Thr7Ala and Ala17Gly in QRDR$^{GyrA}$ to the susceptibility of FQs, the GyrA was edited to alter the Thr to Ala at position 7 in QRDR$^{GyrA}$ (QRDR$^{GyrAT7A}$) and Ala to Gly at position 17 in QRDR$^{GyrA}$ (QRDR$^{GyrAA17G}$) separately. We observed that the Ala17Gly mutation in QRDR$^{GyrA}$ confers hypersensitivity to FQs but not Thr7Ala (Fig. 1C and Table 1). These results imply that Ala17 in QRDR$^{GyrA}$ of mycobacteria plays a key role in susceptibility to FQs and hints that the mycobacterium bearing Ala17Gly or Ala17Ser amino acid substitution in QRDR$^{GyrA}$ may be hypersensitive to FQs.

To the best of our knowledge, this is the first detailed study to underscore the contribution of the amino acid substitutions in GyrA to FQs resistance in mycobacteria using the CRISPR-associated recombintion for gene editing in *M. abscessus*. Our observations are in strong agreement with a previous study in which it was found by peptide sequences alignment that the amino acid at position 17 of QRDR$^{GyrA}$ was likely involved in the intrinsic resistance of mycobacteria to quinolones (9). In addition, although both Ala17Gly and Ala17Ser of QRDR$^{GyrA}$ could cause hypersensitivity of *M. abscesses* to FQs, the susceptibility of MOX to QRDR$^{GyrAA17S}$ and QRDR$^{GyrAA17G}$ is different. Following the Clinical and Laboratory Standards Institute (CLSI) guidelines, the breakpoint for MOX resistance in *M. abscessus* was determined to be 4 µg/mL (15). Consequently, QRDR$^{GyrAA17G}$ is still classified as resistant to MOX. The MICs of FQs to the hypersensitive *M. abscessus* mutants (QRDR$^{GyrAA17S}$ or QRDR$^{GyrAA17G}$) are still higher than that to *M. tuberculosis*, which indicates that besides the contribution of the 17Ala of QRDR$^{GyrA}$ in *M. abscessus*, other factors leading to higher MICs of FQs to *M. abscessus* may exist. A recent study analyzed the *gyrA* and *gyrB* of FQs-resistant *M. abscessus* isolates

**TABLE 1**  MICs of various drugs for different *M. abscessus* strains$^a$

| Antibiotics | *M. abscessus* strains/MICs (µg/mL) | | | | | |
| --- | --- | --- | --- | --- | --- | --- |
| | QRDR$^{WT}$ | QRDR$^{GyrAA17S}$ | QRDR$^{GyrAA17S17A}$ | QRDR$^{GyrAA17G+T7A}$ | QRDR$^{GyrAA17G}$ | QRDR$^{GyrAT7A}$ |
| Levofloxacin | 16 | 4 | 16 | 4 | 4 | 16 |
| Moxifloxacin | 8 | 2 | 8 | 4 | 4 | 8 |
| Rifabutin | 4 | 4 | 4 | 4 | 4 | 4 |
| Amikacin | 4 | 4 | 4 | 4 | 4 | 4 |

$^a$Broth microdilution method was used to determine the MICs. The MIC was defined as the lowest drug concentration that prevented visible bacterial growth. The experiment was performed in triplicate and repeated twice.

but found no mutation in them, which also suggested that besides *gyrA* and *gyrB,* other mechanisms also contribute to FQs resistance in *M. abscessus* (16).

## ACKNOWLEDGMENTS

This work was supported by the National Key R&D Program of China (2021YFA1300904), partially by the National Natural Science Foundation of China (NSFC 81973372, 21920102003), the Joint Research Health Research Council of New Zealand (HRC 20/1211)-NSFC Collaboration grant (82061128001), and the Chinese Academy of Sciences (154144KYSB20190005, YJKYYQ20210026). The funders had no role in study design, data collection and analysis, decision to publish, or preparation of the manuscript. All authors read and approved the final version of the manuscript.

## AUTHOR AFFILIATIONS

[1]State Key Laboratory of Respiratory Disease, Guangzhou Institutes of Biomedicine and Health, Chinese Academy of Sciences, Guangzhou, China
[2]University of Chinese Academy of Sciences, Beijing, China
[3]Guangdong-Hong Kong-Macao Joint Laboratory of Respiratory Infectious Diseases, Guangzhou Institutes of Biomedicine and Health, Chinese Academy of Sciences, Guangzhou, China
[4]China-New Zealand Joint Laboratory on Biomedicine and Health, Guangzhou, China
[5]School of Life Sciences, University of Science and Technology of China, Hefei, Anhui, China
[6]Institutes of Physical Science and Information Technology, Anhui University, Hefei, Anhui, China
[7]State Key Laboratory of Respiratory Disease, Guangzhou Chest Hospital, Guangzhou, China
[8]State Key Laboratory of Respiratory Disease, National Clinical Research Center for Respiratory Disease, The National Center for Respiratory Medicine, The First Affiliated Hospital of Guangzhou Medical University, Guangzhou, China
[9]Department of Microbiology and Immunology, School of Biomedical Sciences, University of Otago, Dunedin, New Zealand
[10]Maurice Wilkins Centre for Molecular Biodiscovery, The University of Auckland, Private Bag, Auckland, New Zealand
[11]National Clinical Research Center for Infectious Diseases, Guangdong Provincial Clinical Research Center for Tuberculosis, Shenzhen Third People's Hospital, Shenzhen, China

## PRESENT ADDRESS

Nanshan Zhong, Guangzhou Laboratory, Bio-Island, Guangzhou, China

## AUTHOR ORCIDs

Shuai Wang http://orcid.org/0000-0003-0893-2534
Jingran Zhang http://orcid.org/0000-0001-9701-4289
H. M. Adnan Hameed http://orcid.org/0000-0003-1993-9547
Tianyu Zhang http://orcid.org/0000-0001-5647-6014

## FUNDING

| Funder | Grant(s) | Author(s) |
| --- | --- | --- |
| MOST \| National Natural Science Foundation of China (NSFC) | NSFC 81973372, 21920102003, 82061128001 | Tianyu Zhang |

## ADDITIONAL FILES

The following material is available online.

## Supplemental Material

**Table S1 (Spectrum02809-23-s0001.docx).** Oligonucleotides used in this study.

## Open Peer Review

**PEER REVIEW HISTORY (review-history.pdf).** An accounting of the reviewer comments and feedback.

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
