## [Reviewer comments · Microbiology Spectrum]

Microbiology Spectrum

Amino acid 17 in QRDR of Gyrase A plays a key role in fluoroquinolones susceptibility in Mycobacteria

Shuai Wang, Jingran Zhang, H.M. Adnan Hameed, Jie Ding, Ping Guan, Xiang Fang, Jiacong Peng, Biyi Su, Shangming Ma, Yaoju Tan, Gregory M. Cook, Guoliang Zhang, Yong-Ping Lin, Nanshan Zhong, Jinxing Hu, Jianxiong Liu, and Tianyu Zhang

Corresponding Author(s): Tianyu Zhang, Guangzhou Institutes of Biomedicine and Health, Chinese Academy of Sciences

Review Timeline:

Submission Date:	July 9, 2023
Editorial Decision:	July 25, 2023
Revision Received:	August 20, 2023
Accepted:	August 27, 2023

Editor: Gyanu Lamichhane

Reviewer(s): The reviewers have opted to remain anonymous.

Transaction Report:

DOI: <https://doi.org/10.1128/spectrum.02809-23>

July 25, 2023

Prof. Tianyu Zhang
Guangzhou Institutes of Biomedicine and Health, Chinese Academy of Sciences
State Key Laboratory of Respiratory Disease
Room 207, 190 Kaiyuan Rd
Science Park
Guangzhou, Guangdong 510530
China

Re: Spectrum02809-23 (Amino acid 17 in QRDR of Gyrase A plays a key role in fluoroquinolones susceptibility in Mycobacteria)

Dear Prof. Tianyu Zhang:

Thank you for submitting your manuscript to Microbiology Spectrum.

Two referees with expertise in mycobacteriology have reviewed your manuscript and provided helpful feedback for revision. In addition to revising the scientific content, it is also my assessment that the manuscript be edited for language use accuracy and clarity.

Link Not Available

Sincerely,

Gyanu Lamichhane

Journals Department
Reviewer comments:

Reviewer #1 (Comments for the Author):

The authors experimentally addressed the role of specific amino acid residues within GyrA of *Mycobacterium abscessus* in fluorochinolone susceptibility. They altered amino acids associated with resistance in *M. tuberculosis* and investigated the phenotype of the recombinant strains. Besides single amino acid alterations, they also introduced double amino acid alterations.

Alterations at position 92 from A to S or G increased susceptibility, a phenotype which was reverted upon re-introduction of the original A. Susceptibility towards unrelated drugs was not affected. Although interesting, the manuscript is difficult to read. The figure from the supplement should be part of the main text, numbers of the amino acids should be indicated above the alignment (please check carefully with the numbers given in the legend). Alterations introduced and corresponding phenotypes should be indicated above the alignment. Which are the most prominent clinically acquired fluoroquinolone conferring alterations in *M. tuberculosis*. What are the predictions for other Mycobacteria. The two different numberings total positions and QRDR makes reading difficult.

Reviewer #2 (Comments for the Author):

In this paper, the researchers verified that gene-edited *M. abscessus* mutants bearing Ser/Gly at amino acid 17 of QRDRGyrA were more susceptible to fluoroquinolones than their parent strain and the revertant, which supports mycobacteria containing Ser/Gly at this position were more susceptible to fluoroquinolones than those containing Ala. This study outlines its scope and focus very clearly. It is well written and easy to understand. The results are presented in a simple-to-understand manner.

Here are some recommendations for the authors' consideration.

A) Lines 108-111: The researchers found that Ala17 in QRDRGyrA of *M. abscessus* plays a potential role in susceptibility to FQs, but the antibiotic resistance mechanism of *M. abscessus* is quite different from *M. tuberculosis*. Besides, *M. smegmatis* rather than *M. abscessus* is commonly used as the model organism of *M. tuberculosis*. Therefore, it is inappropriate to conclude that "clinical *M. tuberculosis* isolates containing Ala17Gly or Ala17Ser amino acid substitutions in QRDRGyrA may be preferentially treated with regimens containing FQs".

B) Line 189, Table 1: The MIC of MOX to the MabGyrAA92G (4 mg/L) were 1/2 of that of MabWT (8 mg/L), while the MIC of MOX to the MabGyrAA92G (2 mg/L) were 1/4 of that of MabWT (8 mg/L). According to the CLSI guideline, the moxifloxacin resistance breakpoint of *M. abscessus* was 4 mg/L, thus MabGyrAA92G and MabWT were both resistant to moxifloxacin. There might be differences in fluoroquinolones susceptibility in *M. abscessus* mutants bearing Ser and Gly at this position and this is recommended for discussion.

Staff Comments:

Preparing Revision Guidelines

Please return the manuscript within 60 days; if you cannot complete the modification within this time period, please contact me. If you do not wish to modify the manuscript and prefer to submit it to another journal, please notify me of your decision immediately so that the manuscript may be formally withdrawn from consideration by Microbiology Spectrum.

Title: Amino acid 17 in QRDR of Gyrase A plays a key role in fluoroquinolones susceptibility in Mycobacteria

ID: Spectrum02809-23

20th August, 2023

Dear Editor and Reviewers:

Thank you for your time and effort in providing us valuable suggestions to improve our manuscript entitled “Amino acid 17 in QRDR of Gyrase A plays a key role in fluoroquinolones susceptibility in Mycobacteria” (ID: Spectrum02809-23). We have carefully addressed all the comments and made corrections accordingly, so now we hope that the modified manuscript can finally be considered for publication in your esteemed journal.

The main corrections are marked in red in the revised manuscript. Responses to the reviewer’s comments are as follows:

Reviewer #1 (Comments for the Author):

The authors experimentally addressed the role of specific amino acid residues within GyrA of *Mycobacterium abscessus* in fluoroquinolone susceptibility. They altered amino acids associated with resistance in *M. tuberculosis* and investigated the phenotype of the recombinant strains. Besides single amino acid alterations, they also introduced double amino acid alterations. Alterations at position 92 from A to S or G increased susceptibility, a phenotype which was reverted upon re-introduction of the original A. Susceptibility towards unrelated drugs was not affected. Although interesting, the manuscript is difficult to read. The figure from the supplement should be part of the main text, numbers of the amino acids should be indicated above the alignment (please check carefully with the numbers given in the legend). Alterations introduced and corresponding phenotypes should be indicated above the alignment. Which are the most prominent clinically acquired fluoroquinolone conferring alterations in *M. tuberculosis*. What are the predictions for other Mycobacteria. The two different numberings total positions and QRDR makes reading difficult.

A) The figure from the supplement should be part of the main text, numbers of the amino acids should be indicated above the alignment (please check carefully with the numbers given in the legend). Alterations introduced and corresponding phenotypes should be indicated above the alignment.

Authors' response:

First of all, we are thankful to the reviewer for your clear understanding and describing the precise theme of our study. Thanks for your suggestion. Figure S1 is now moved to the main text of the revised manuscript which is now marked as 'Figure 1A'. Alterations introduced and corresponding phenotypes have been indicated above the alignment in 'Figure 1A'. We have modified the figure and its legend.

B) Which are the most prominent clinically acquired fluoroquinolone conferring alterations in *M. tuberculosis*. What are the predictions for other Mycobacteria.

Authors' response: We acknowledge your concern, in fact, only A17G of QRDR^{GyrA} conferring fluoroquinolone hypersensitivity was found in clinical *M. tuberculosis* isolates. For other Mycobacteria, we hypothesized that A17S or A17G of QRDR^{GyrA} could cause hypersensitivity to fluoroquinolones. Therefore, the *Mycobacterium peregrinum*, *Mycobacterium fortuitum* and *Mycobacterium aurum* are more susceptible to fluoroquinolones than others because of the presence of a Ser at position 17 of QRDR^{GyrA} in these three mycobacteria. We have also described this in the revised manuscript (Lines 117-118).

C) The two different numberings total positions and QRDR makes reading difficult.

Authors' response: Thank you for pointing out this problem in manuscript. We have revised the numbering of the full text and uniformly use the QRDR^{GyrA} to indicate the editing sites in the revised manuscript.

Reviewer #2 (Comments for the Author):

In this paper, the researchers verified that gene-edited *M. abscessus* mutants bearing Ser/Gly at amino acid 17 of QRDR^{GyrA} were more susceptible to fluoroquinolones than their parent strain and the revertant, which supports mycobacteria containing Ser/Gly at this position were more susceptible to fluoroquinolones than those containing Ala.

This study outlines its scope and focus very clearly. It is well written and easy to understand. The results are presented in a simple-to-understand manner.

Here are some recommendations for the authors' consideration.

A) Lines 108-111: The researchers found that Ala17 in QRDR^{GyrA} of *M. abscessus* plays a potential role in susceptibility to FQs, but the antibiotic resistance mechanism of *M. abscessus* is quite different from *M. tuberculosis*. Besides, *M. smegmatis* rather than *M. abscessus* is commonly used as the model organism of *M. tuberculosis*. Therefore, it is inappropriate to conclude that "clinical *M. tuberculosis* isolates containing Ala17Gly or Ala17Ser amino acid substitutions in QRDR^{GyrA} may be preferentially treated with regimens containing FQs".

Authors' response: We are grateful to the reviewer for appreciating our work and thanks for your productive recommendations for improving our manuscript. We have modified the concerned text section in the revised manuscript (Lines 117-118).

B) Line189, Table 1: The MIC of MOX to the Mab^{GyrAA92G} (4 mg/L) was 1/2 of that of Mab^{WT} (8 mg/L), while the MIC of MOX to the Mab^{GyrAA92S} (2 mg/L) was 1/4 of that of Mab^{WT} (8 mg/L). According to the CLSI guideline, the moxifloxacin resistance breakpoint of *M. abscessus* was 4 mg/L, thus Mab^{GyrAA92G} and Mab^{WT} were both resistant to moxifloxacin. There might be differences in fluoroquinolones susceptibility in *M. abscessus* mutants bearing Ser and Gly at this position and this is recommended for discussion.

Authors' response: We gratefully appreciate your valuable suggestion. We have also described this in the discussion section of the revised manuscript (Lines 124-129).

August 27, 2023

Prof. Tianyu Zhang
Guangzhou Institutes of Biomedicine and Health, Chinese Academy of Sciences
State Key Laboratory of Respiratory Disease
Room 207, 190 Kaiyuan Rd
Science Park
Guangzhou, Guangdong 510530
China

Re: Spectrum02809-23R1 (Amino acid 17 in QRDR of Gyrase A plays a key role in fluoroquinolones susceptibility in Mycobacteria)

Dear Prof. Tianyu Zhang:

Your manuscript has been accepted, and I am forwarding it to the ASM Journals Department for publication. You will be notified when your proofs are ready to be viewed.

Sincerely,

Gyanu Lamichhane
Editor, Microbiology Spectrum
